# CF-GISS: Collision-Free Generative 3D Indoor Scene Synthesis with Controllable Floor Plans and Optimized Layouts

## Abstract

We introduce CF-GISS, a novel framework for generative 3D indoor scene synthesis that ensures *collision-free* scene layouts by incorporating an image-based intermediate layout representation. In contrast to existing methods that directly construct the scene graph or object list, our approach facilitates substantially more effective prevention of collision artifacts as *out-of-distribution* (OOD) scenarios during generation. Furthermore, CF-GISS conditions layout generation on floor plans *controllable* via images or textual descriptions, enabling the production of coherent, house-wide layouts that are robust to variations in geometric and semantic structures. Our framework demonstrates *state-of-the-art* performance on the 3D-FRONT dataset, delivering high-quality, collision-free scene synthesis while offering flexibility in accommodating a range of floor plan structures. Additionally, we propose a *novel dataset* with significantly expanded coverage of household items and room configurations, as well as improved data quality.

## 1 Introduction

Generative 3D indoor scene synthesis (Xu et al., 2002; Merrell et al., 2011; Yu et al., 2011; Fisher et al., 2012; Qi et al., 2018; Zhang et al., 2018; Li et al., 2018a; Ritchie et al., 2019; Wang et al., 2019; Yao et al., 2024; Min et al., 2024; Vaswani et al., 2017; Hu et al., 2020; Wang et al., 2020; Paschalidou et al., 2021; Leimer et al., 2022; Tang et al., 2024; Lin & Mu, 2024) is essential not only for creative and technical workflows such as interior design, architectural planning, video game development, and virtual or augmented reality, but also for advancing embodied AI by providing simulated environments for training. This approach facilitates rapid prototyping, reduces manual labor, lowers deployment costs, and accelerates iteration. Despite advances in neural representations such as NeRFs (Mildenhall et al., 2020) and Gaussian splatting (Kerbl et al., 2023), mesh-based assets remain the predominant 3D representation in these applications due to their superior rendering quality and explicit interactivity, particularly in video games, design, and simulation. Consequently, most generative indoor scene synthesis methods (Zhang et al., 2018; Li et al., 2018a; Ritchie et al., 2019; Wang et al., 2019; Paschalidou et al., 2021; Tang et al., 2024; Lin & Mu, 2024) follow a procedural generation paradigm, constructing a scene graph or object list for the scene layout, with each node containing detailed specifications for individual scene objects, which can be subsequently retrieved from a CAD asset dataset and rendered by various graphics engines.

However, a fundamental limitation of directly constructing scene graphs or object lists is the *inability* to prevent implausible collisions or overlaps between objects, such as a bed intersecting with cabinets, during the generation process. While collision detection can be performed as a post-processing step, it is computationally expensive, and resolving the detected collisions during post-processing remains nontrivial.

In this paper, we propose CF-GISS, a novel framework aimed at resolving the issue of *strictly preventing collision artifacts* in generative 3D indoor scene synthesis during the generation process. Central to our approach is the synthesis of an RGB image representing the scene layout as an intermediate step in the procedural workflow, which is subsequently converted into an object list or graph, rather than directly constructing the graph in a single step. Our key insight is that, compared to graph representations, which are inherently tabular, introducing an RGB image as an intermediate

representation enables more effective detection of collision artifacts as *out-of-distribution* (OOD) scenarios. Such collisions are characterized by a lower likelihood of occurrence during training, leading to a large loss value, which ultimately facilitates strict avoidance of collisions during the generation process without the need for post-processing. Our extensive evaluation of CF-GISS on the 3D-FRONT dataset (Fu et al., 2021a) demonstrates its *state-of-the-art* performance both qualitatively and quantitatively, particularly in the near elimination of unreasonable object collisions, a prevalent issue in existing methods.

Meanwhile, the floor plan structure serves as a crucial yet often overlooked constraint in recent works on generative indoor scene synthesis, as room sizes, along with the placement of doors and windows, substantially influence the logical arrangement of household layouts. Unlike methods that focus on generating single-room layouts (Zhang et al., 2018; Li et al., 2018a; Ritchie et al., 2019; Wang et al., 2019; Paschalidou et al., 2021; Tang et al., 2024; Lin & Mu, 2024), CF-GISS is conditioned on the overall floor plan structure, which can be controlled via either images or text prompts, to synthesize coherent, house-wide indoor scenes adaptable to floor plan variations in geometric and semantic structures.

Finally, we introduce a large-scale dataset comprising 9,706 scenes with floor plans and scene layouts, approximately 1.4 times larger than 3D-FRONT. Our dataset expands household item coverage to 26 super-categories, including items from kitchens, bathrooms, and balconies, addressing gaps in 3D-FRONT, which lacks furnishings in these areas and occasionally leaves living rooms or bedrooms unfurnished. It also resolves common issues found in 3D-FRONT, such as misclassified objects, unrealistic placements, and collisions, providing clean layouts without requiring extensive data cleaning. We demonstrate applications of this new dataset using CF-GISS.

Our contributions are summarized as: *i)* **Novel framework** - a novel framework for *collision-free* generative 3D indoor scene synthesis with *controllable* floor plans via images or text prompts, optimized room layouts, and photorealistic rendering; *ii)* **Performance** - state-of-the-art performance on the 3D-FRONT dataset, both qualitatively and quantitatively, especially the near elimination of unreasonable object collisions; and *iii)* **Dataset**[1] - a novel dataset of 9,706 scenes with floor plans and scene layouts, 1.4 times larger than 3D-FRONT, with improved item coverage and data quality.

## 2 RELATED WORK

Early work employed a rule-based constraint satisfaction formulation to generate 3D room layouts for pre-specified sets of objects (Xu et al., 2002). Other approaches optimized cost functions based on interior design principles (Merrell et al., 2011) and object-object statistical relationships (Yu et al., 2011). The earliest data-driven approach modeled object co-occurrences using a Bayesian network and Gaussian mixtures to capture pairwise spatial relations extracted from 3D scenes (Fisher et al., 2012). With the availability of large datasets of 3D environments, such as SUNCG (Song et al., 2017), 3D-FRONT (Fu et al., 2021a), SUN3D (Xiao et al., 2013), Matterport3D (Chang et al., 2017), InteriorNet (Li et al., 2018b), Structured3D (Zheng et al., 2020), and 3D-FURNITURE (Fu et al., 2021b), learning-based approaches have gained popularity. Various methods for indoor scene synthesis have been proposed, including: human-centric probabilistic grammars (Qi et al., 2018), Generative Adversarial Networks (GANs) trained on matrix representations of scene objects (Zhang et al., 2018), recursive neural networks for sampling 3D scene hierarchies (Li et al., 2018a), convolutional neural networks (CNNs) trained on top-down room images (Zhang et al., 2018; Ritchie et al., 2019), spatial prior graph neural networks trained on labeled 3D spatial relationships (Wang et al., 2019; Yao et al., 2024), and Variational Autoencoder (VAE) models trained on top-down functional furniture group images (Min et al., 2024). Additionally, floor plan synthesis approaches have been proposed using graph neural networks (Hu et al., 2020). With the development of transformer models (Vaswani et al., 2017), transformer-based approaches have become increasingly popular. These include floor plan-conditioned furniture synthesis and text-conditioned furniture synthesis using transformer autoregressive models (Wang et al., 2020; Paschalidou et al., 2021), as well as methods integrating expert knowledge in the form of differentiable scalar functions to guide the generation of more ergonomic layouts (Leimer et al., 2022). Recently, diffusion models (Ho et al., 2020) have demonstrated impressive visual quality in generative tasks, including indoor furniture synthesis (Tang et al., 2024; Lin & Mu, 2024). However, these methods primarily synthesize layouts

---

[1]The dataset will be publicly released upon acceptance.

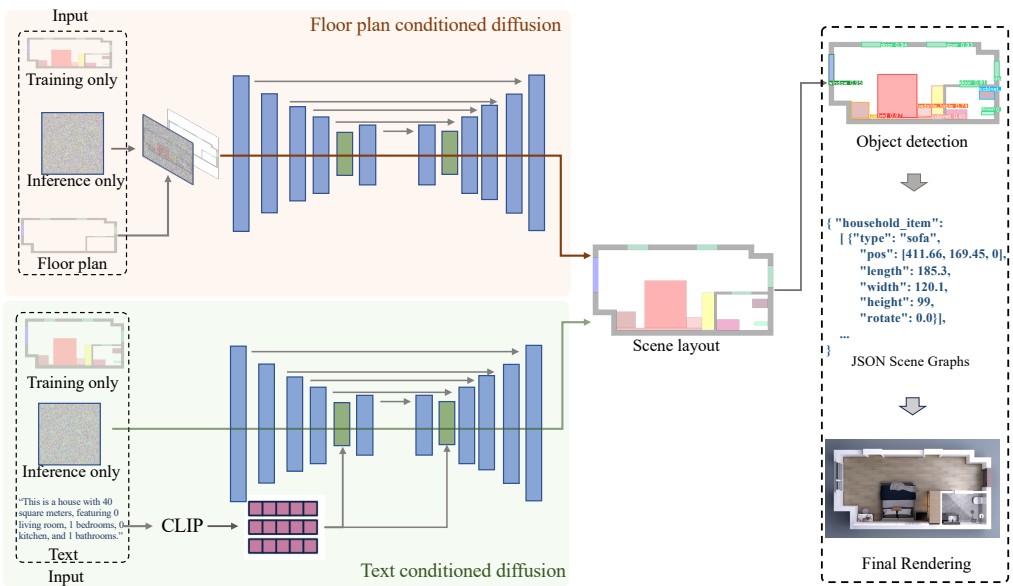

Figure 1: Overview of CF-GISS. First, we generate the scene layout using a conditional diffusion model, conditioned either on a floor plan image or a text description. Next, we apply object detection to identify individual household items and use a structured scene graph to hierarchically organize the spatial relationships between rooms and objects, along with their attributes. Finally, the scene is rendered into photorealistic images.

for individual rooms rather than house-wide scenes that consider the overall floor plan structure. Additionally, unlike recent works such as Tang et al. (2024), Lin & Mu (2024), and Yang et al. (2024), which directly synthesize the scene graph using a 1D-Unet, our approach employs a 2D-Unet. This enables a better understanding of the spatial relationships between doors, windows, and furniture, thereby more effectively preventing collisions, doorway blockages, and other artifacts. Such artifacts are also present in widely-used indoor scene datasets, such as 3D-FRONT (Fu et al., 2021a;b), which requires substantial effort for data cleaning.

## 3 METHOD

We propose a novel pipeline for 3D-aware indoor scene synthesis, as depicted in Figure 1. The pipeline starts with a floor plan description—provided either as an image or text—and use a conditional diffusion model to generate a corresponding 2D scene layout. The use of this 2D representation enables us to leverage efficient image encoders for layout generation while effectively preventing object overlaps. Next, we employ automatic object detectors and segmentation maps to identify individual household items and extract a structured scene graph that hierarchically organizes spatial relationships and object attributes. Finally, the 3D scene objects are retrieved accordingly and rendered to produce photorealistic 3D-consistent images.

### 3.1 DIFFUSION-BASED SCENE LAYOUT GENERATION

**Image-conditioned floor plans** We leverage the recent success of image-based diffusion models (Saharia et al., 2022a; Rombach et al., 2022; Amit et al., 2021) and frame the problem of generating diverse, realistic indoor scene layouts as a conditional image-to-image translation task, as illustrated in Figure 1 (top-left). Unlike complex scene graphs or tabular formats, natural RGB images serve as a convenient intermediate representation for the layout, easily processed by existing vision tools. Crucially, since RGB images are easy-to-interpret by an appropriate encoder, we can construct a highly effective conditional generative model that accurately captures the data distribution. In RGB images, collisions between objects are immediately visible and flagged as OOD samples, allowing the model to generate coherent, physically realistic layouts.

| Item | Color | Item | Color | Item | Color |
|------|-------|------|-------|------|-------|
| Bed | FF0000 | Cabinet | FFFF00 | Bed Background | FF3333 |
| Bedside Table | F08080 | Table | A52A2A | Leisure Sofa | 666600 |
| Sofa | FF9933 | TV Cabinet | FFCC99 | Sofa Background | 99004C |
| Coffea Table | CCFF99 | Dining Cabinet | FF9999 | Shoe Cabinet | 006633 |
| Single Sofa | CC6600 | Dining Table | FF6666 | Side Coffea Table | 99FFCC |
| Single Door Floor Cabinet | 9999FF | Double Door Floor Cabinet | 6666FF | Cooker Cabinet | 000099 |
| Sink Cabinet | 0000CC | Electrical FLoor Cabinet | 3333FF | Refrigerator | 006666 |
| Shower | 33FF99 | Toilet | 660033 | Washbasin | CC0066 |
| Washing Machine | FFCCE5 | Washing Set | FF66B2 | | |
| Wall | 000000 | Door | 139C5A | Window | 0000FF |

Table 1: Scene layout items and corresponding colors, with the opacity level set to 0.3.

Specifically, given an image of an empty floor plan $\boldsymbol{y}$, we train a diffusion model $\epsilon_\theta(\boldsymbol{x}; \boldsymbol{y_i}, t)$ to model the conditional distribution of the corresponding layouts $p(\boldsymbol{x} \mid \boldsymbol{y_i})$ [2], where $\epsilon_\theta$ is structured as a 2D U-Net, following (Ho et al., 2020), with 3 input channels (random noise) and 3 output channels (the predicted layout image). To incorporate floor plan image conditioning, we expand the U-Net input from 3 to 6 channels. During training, a predetermined noise schedule realizes a Markov chain, yielding the diffused sample $\boldsymbol{x_t}(\boldsymbol{x}, \boldsymbol{y_i}, t, \epsilon)$, where $\epsilon \sim \mathcal{N}(\boldsymbol{0}, \boldsymbol{I})$ and $t \sim \mathcal{U}(0, 1)$. The loss function is given by the denoising score matching objective (Ho et al., 2020):

$$\mathbb{E}_{(\boldsymbol{x}, \boldsymbol{y_i}) \sim p_{\text{data}}, \epsilon \sim \mathcal{N}(\boldsymbol{0}, \boldsymbol{I}), t \sim \mathcal{U}(0,1)} \left[ \|\epsilon_\theta(\boldsymbol{x}; \boldsymbol{y_i}, t) - \epsilon\|^2 \right]. \tag{1}$$

**Text-conditioned floor plans**  Another convenient method to specify the floor plan is through natural language, particularly in practical scenarios where floor plan images are unavailable or incompatible with the format accepted by our model. Given the success of text-to-image diffusion models (Ramesh et al., 2022; Saharia et al., 2022b) and our use of RGB images to represent filled layouts, text descriptions are a viable alternative within our framework. In this case, we generate a fixed-size conditioning vector $\boldsymbol{y_c}$ by passing the text input through a CLIP encoder (Radford et al., 2021). Semi-structured text is particularly effective for such a task, for example, "This is a flat with 40 square meters, featuring 0 living rooms, 1 bedroom, 0 kitchens, and 1 bathroom". The resulting CLIP embedding serves as the conditional variable for the diffusion model, guiding the generation of furniture layouts based on high-level semantic information encoded in the text description. This allows for more intuitive control over the layout generation by leveraging natural language as an additional input modality.

The text-based model is trained with the same procedure, using Equation 1 but the conditioning variable is the CLIP embedding $\boldsymbol{y_c}$ instead of the empty floor plan $\boldsymbol{y_i}$. Another minor difference is that conditioning is introduced through a cross-attention layer (Vaswani, 2017) near the UNet bottleneck, as illustrated in Figure 1 (bottom-left). Although the CLIP was designed to handle both image and text conditioning, we opt for operating directly in pixel space for image conditioning to ensure higher-resolution image generation and preserve finer details, mitigating the compression artifacts found in latent diffusion, albeit at a higher computational cost. Nevertheless, with separate encoders CF-GISS seamlessly handles both image and text inputs, catering to different user preferences.

### 3.2 HIERARCHICAL SCENE GRAPH EXTRACTION AND OBJECT RETRIEVAL

To generate a scene graph from the candidate layout $\boldsymbol{x} \sim p(\boldsymbol{x} \mid \boldsymbol{y})$, we follow the automatic framework proposed by (Lv et al., 2021). As depicted in Figure 2, we start by fine-tuning YOLOv8 (Jocher et al., 2023) to detect the locations and attributes of all objects present in $\boldsymbol{x}$. The color of each object uniquely identifies its category from a set of 28 household item categories and 3 floor plan item categories, as listed in Table 1. The other relevant attributes are then populated according to predetermined rules to produce an object list $\mathcal{O} = (\boldsymbol{o_1}, \boldsymbol{o_2}, \ldots, \boldsymbol{o_n})$. This facilitates the automated extraction of object properties such as position, orientation, and size based on predefined standards. We employ YOLOv8 to simultaneously obtain the segmentation maps for each category of rooms, including living rooms, bedrooms, kitchens, bathrooms, and balconies.

---

[2]The subscript on $\boldsymbol{y}$ differentiates image conditioning $\boldsymbol{y_i}$ from natural language conditioning via CLIP $\boldsymbol{y_c}$

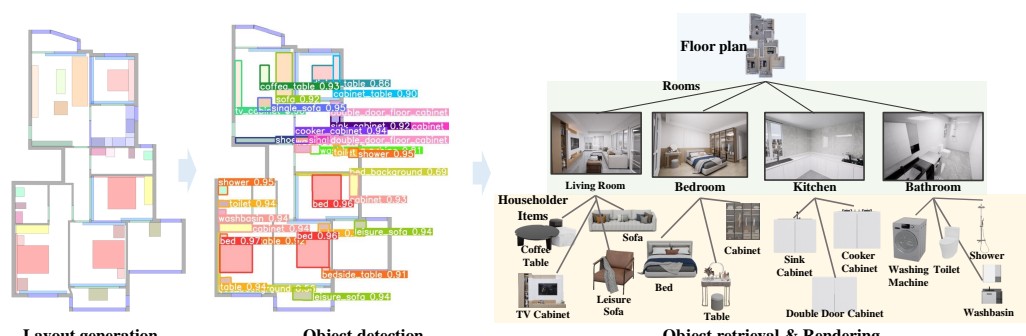

Figure 2: Scene graph extraction and object retrieval.

Given the dimensions and category of each object, we deterministically retrieve an example from a category-specific textured mesh database $\mathcal{D}$ such that it has the smallest size difference:

$$\boldsymbol{e_i} = \arg\min_{\boldsymbol{e} \in \mathcal{D}}(\|o_i^x - e_i^x\|^2 + \|o_i^y - e^y\|^2) : o_i^c = e^c, \forall \boldsymbol{o_i} \in \mathcal{O}. \tag{2}$$

The set of retrieved examples $\{\boldsymbol{e_1}, \boldsymbol{e_2}, \ldots, \boldsymbol{e_n}\}$ constitutes the leaf nodes of the scene graph; see third row in Figure 2 (right). To position the objects in each room and construct the hierarchal scene graph, we utilize the semantic detection and segmentation outputs of household items and rooms from YOLOv8. We straighten the edges of the room polygons, similar to (Lv et al., 2021), to reduce uneven lines, and attach doors and windows to these edges, ensuring corrected wall positions that enclose the room.

## 3.3 RENDERING

Finally, we convert the structured scene graph into a 3D mesh. The wall and floor materials for each $\boldsymbol{e_i}$ are procedurally sampled, while being aware of the rooms to which they belong. To maintain uniform lighting and shadow consistency across the scene, an appropriately sized area light is placed at the center of each room. The UE engine (Epic Games) is subsequently utilized to generate photorealistic renderings.

The key advantage of the multi-stage pipeline of CF-GISS (see Figure 1), which first generates a 2D layout rather than directly synthesizing an object scene graph (Tang et al., 2024; Lin & Mu, 2024), lies in its inherent ability to prevent object intersections. By leveraging the 2D RGB images as an intermediate representation, our approach ensures that the spatial relationships between household items are preserved, avoiding common issues such as object overlap and collision, which we validate in Section 5.1.

## 4 DATASET

We collected a large-scale dataset, which we refer to as CF-dataset, of indoor scenes with floor plans and scene layouts, comprising a total of 9,706 different design schemes, approximately 1.4 times larger than the 3D-FRONT dataset (Fu et al., 2021a). This dataset was meticulously created by professional interior designers, consisting of vectorized data stored in the JSON format, as exampled in Appendix List 1, including wall lines, doors, windows, and household items such as furniture, fixtures, and appliances. While this dataset primarily focuses on optimal layouts and is currently linked to a limited pool of CAD models, users are free to retrieve assets from any public dataset (3D66, 2013; Fu et al., 2021b). The data description is as follows:

- **Rooms**: Represented as enclosed loops of interior wall lines, defined by 2D coordinates.
- **Doors, Windows, and Household Items**: Represented as 2D bounding boxes, defined by category, 3D coordinates, orientation, and dimensions (length, width, height).

Our CF-dataset offers several clear advantages over the 3D-FRONT dataset:

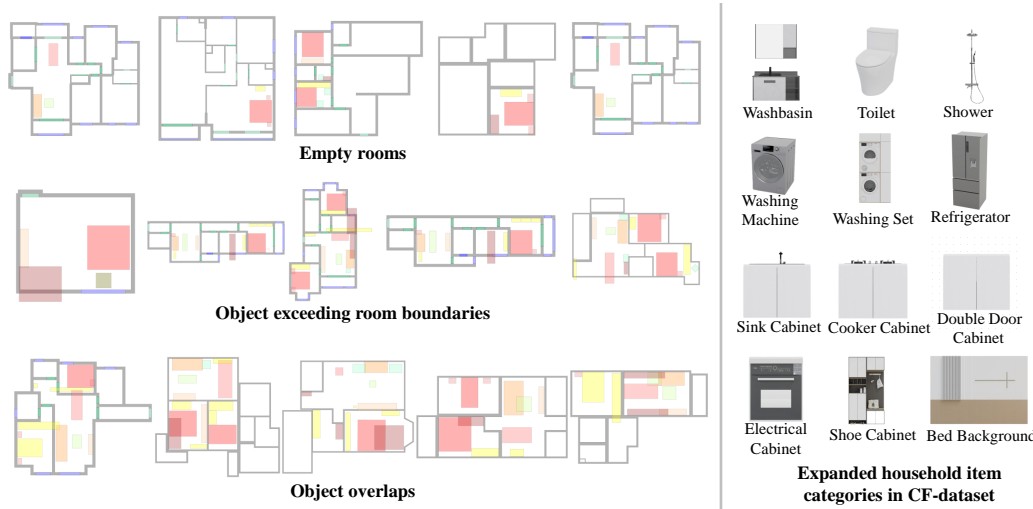

Figure 3: **Left** - Erroneous scenes in 3D-FRONT. **Right** - Expanded categories in CF-dataset.

**Expanded Coverage of Household Items and Room Categories** While 3D-FRONT provides instance semantic labels for 34 categories and 10 super-categories of household items, its dataset primarily includes items placed in living rooms, bedrooms, and dining rooms, with no items for kitchens, bathrooms, or balconies. Consequently, the layouts in 3D-FRONT are consistently devoid of furnishings in these areas. Furthermore, there are instances in 3D-FRONT where living rooms or bedrooms are unnaturally left unfurnished. Our CF-dataset fills this gap by offering 26 super-categories of household items, including furniture, fixtures, and appliances, that comprehensively cover living rooms, bedrooms, dining rooms, kitchens, bathrooms, and balconies. As illustrated in Figure 3, our CF-dataset not only contains more valid living rooms and bedrooms (each with at least one household item in place), but also includes outfitted kitchens and bathrooms. A comprehensive statistic of our CF-dataset in comparison with 3D-FRONT in terms of household item super-categories and their occurrences in layouts is detailed in Figure 4 and Appendix Table 3.

**Improved Data Quality** As reported by Zhang et al. (2018); Ritchie et al. (2019); Paschalidou et al. (2021); Tang et al. (2024); Lin & Mu (2024), the 3D-FRONT dataset contains erroneous layouts with unnatural object sizes, misclassified items, and unrealistic object placements (*e.g.*, furniture outside room boundaries, lamps on the floor, blockage of doorways, and overlapping objects). Consequently, previous work (Zhang et al., 2018; Ritchie et al., 2019; Paschalidou et al., 2021; Tang et al., 2024; Lin & Mu, 2024) using 3D-FRONT for training and evaluation invested considerable effort in data cleaning, removing numerous layouts with artifacts, which greatly reduced the amount of valid data. For example, after performing a similar filtering process ourselves for 3D-FRONT, only 4,847 valid houses remained out of 6,813 layouts. In contrast, our dataset eliminates these artifacts, as illustrated in Figure 3 and Table 4, as well as Appendix Table 4 and Table 5.

## 5 EXPERIMENTS

We conducted three experiments to assess the performance of CF-GISS and the CF-dataset. First, we trained CF-GISS with floor plan image conditioning on the 3D-FRONT dataset and compared its performance with prior work evaluated also on 3D-FRONT. We particularly evaluate the effectiveness of CF-GISS in preventing collision artifacts by assessing its ability to identify these scenarios as out-of-distribution samples. Next, we trained CF-GISS using both floor plan image conditioning and text conditioning on our CF-dataset to demonstrate its versatility across various use cases.

### 5.1 EVALUATION ON 3D-FRONT

**Dataset and Implementation** We trained CF-GISS with floor plan image conditioning on the 3D-FRONT dataset, consisting of 6,813 scenes, of which 4,847 were retained after a cleaning process. This process excluded layouts lacking furniture, containing furniture exceeding room boundaries,

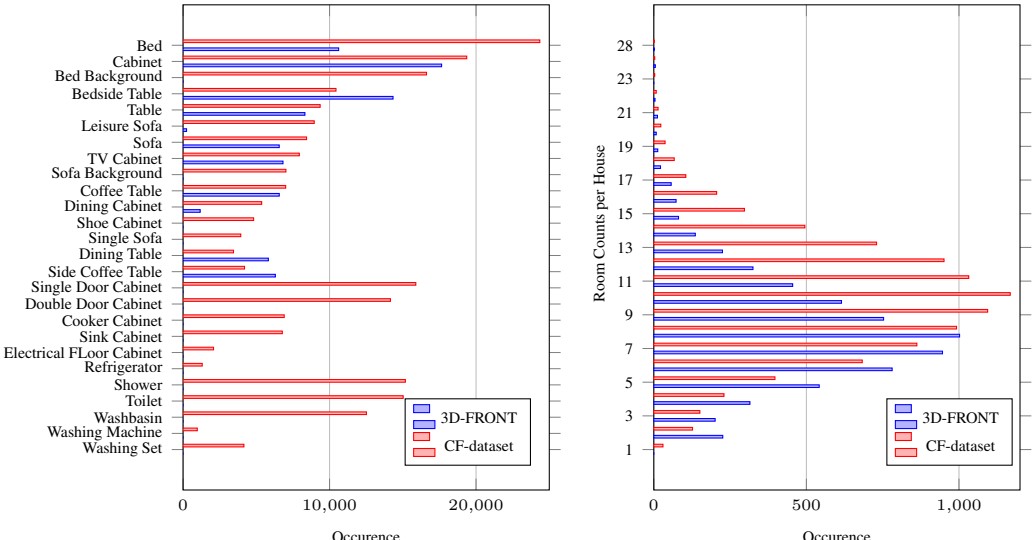

(a) Distribution of household item occurrences per super-category.

(b) Distribution of room counts per house, with an average of 9.78 and a total of 94,964 counts.

Figure 4: Statistics of CF-dataset (ours) in comparison with 3D-FRONT.

or exhibiting collisions. We used 80% of the dataset for training and 20% for testing. Training was conducted on four RTX8000 GPUs with a batch size of 4 for 400 epochs. The initial learning rate is set to 1e-4, with a decay factor of 0.1 applied every 100 epochs. For the diffusion process, we adhere to the default configuration of DDPM, where noise intensity gradually increases from 0 to 1 across 1000 time steps.

**Competitors** We compare CF-GISS with the latest work DiffuScene (Tang et al., 2024), InstructScene (Lin & Mu, 2024), and PhyScene (Yang et al., 2024), which also aim to optimize indoor scene layouts. We used the checkpoint from the DiffuScene unconditional model to generate top-down views of furniture arrangements in bedrooms and living rooms, with a resolution of $256 \times 256$. For InstructScene, we similarly used the checkpoint from the unconditional model to generate bedroom views at the same resolution. However, InstructScene did not release unconditional model checkpoints for living rooms. For PhyScene, we used the checkpoint from the floorplan-conditioned model to generate living room layouts at the same resolution. PhyScene did not release model checkpoints for bedrooms. To ensure fairness in the comparison, the furniture categories generated by DiffuScene, InstructScene, and PhyScene were mapped to our categories as detailed in Appendix List 2. Note that DiffuScene, InstructScene, and PhyScene are *unable* to synthesize house-wide layouts but individual categories of rooms.

| | Dataset | Bedroom | | | | Living Room | | | | Entire House | | | |
|---|---|---|---|---|---|---|---|---|---|---|---|---|---|
| | | FID↓ | KID↓ | POR↓ | PIoU↓ | FID↓ | KID↓ | POR↓ | PIoU↓ | FID↓ | KID↓ | POR↓ | PIoU↓ |
| DiffuScene | 3D-FRONT | 15.91 | 0.04 | 0.1632 | 0.0152 | 45.89 | 0.034 | 0.05 | 0.012 | - | - | - | - |
| InstructScene | 3D-FRONT | 22.35 | 0.02 | 0.2039 | 0.0088 | - | - | - | - | - | - | - | - |
| PhyScene | 3D-FRONT | - | - | - | - | 117.29 | 0.119 | 0.389 | 0.0134 | - | - | - | - |
| CF-GISS (ours) | 3D-FRONT | **14.78** | **0.008** | **0.0766** | **0.0013** | **24.15** | **0.018** | 0.0207 | **0.0015** | **11.51** | **0.01** | 0.0130 | **0.0005** |
| CF-GISS (ours) | CF-dataset | 21.86 | 0.0157 | 0.1049 | 0.0025 | 48.24 | 0.04 | **0.0179** | 0.0029 | 29.97 | 0.039 | **0.0125** | 0.0007 |

Table 2: Quantitative evaluation of CF-GISS, demonstrating superior performance across all metrics compared to prior approaches.

**Results** We present the qualitative evaluation in Figure 5. CF-GISS effectively synthesizes diverse and plausible collision-free household layouts, while both DiffuScene and InstructScene produce a significant portion of layouts with implausible collisions, making them less suitable for practical applications. We present the quantitative evaluation in Table 2. Following previous work, we employ Frechet Inception Distance (FID) (Heusel et al., 2017) and Kernel Inception Distance (KID)

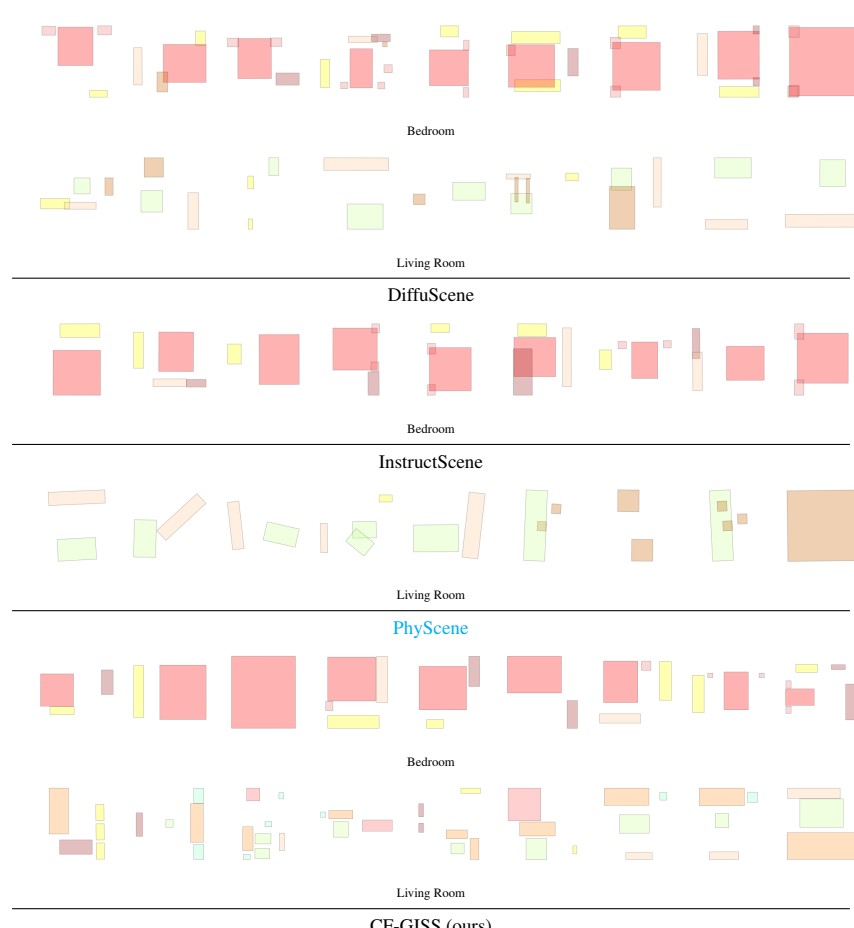

Figure 5: Visualization of synthesized layouts by CF-GISS, DiffuScene (Tang et al., 2024), InstructScene (Lin & Mu, 2024), and PhyScene (Yang et al., 2024), all trained on 3D-FRONT, using the color scheme described in Table 1. All results were randomly selected from an arbitrary batch *without any cherry-picking*. It is evident that only CF-GISS produces clean, collision-free layouts, whereas other methods exhibit a significant number of implausible overlapping items.

(Bińkowski et al., 2018) to evaluate the quality and diversity of the synthesized layouts images. We additionally compute two metrics to evaluate the collisions of 2D bounding boxes in synthesized layouts: the Pairwise Overlap Ratio (POR), which represents the ratio of intersecting furniture pairs to the total number of furniture pairs, and the Pairwise Intersection over Union (PIoU), which measures the ratio of the intersecting area between two furniture items to the union of their areas. The average values for these metrics are calculated by first determining the value for each scene and then computing their arithmetic mean.

**Collision as OOD Samples**   In diffusion models, the training loss is computed as the reconstruction error of the data given the noise, serving as an approximation of the negative log-likelihood (NLL). Assuming the model has been adequately trained on the data distribution, a sample with a high training loss indicates a high NLL, pointing to a potential out-of-distribution (OOD) scenario within the learned distribution that is *improbable* to be generated during sampling. To demonstrate the effectiveness of CF-GISS in preventing collision artifacts as OOD samples, we computed the training loss for clean 3D-FRONT layout samples devoid of collisions and for a set of 400 3D-FRONT samples exhibiting the largest PIoU values. The loss was calculated by adding noise to the samples at timesteps ranging from 900 to 1000, measuring the mean squared error between the true and predicted noise, and averaging the results over 100 iterations. The results indicate an average loss of $5.37 \times 10^{-5}$ for the clean samples and $7.10 \times 10^{-5}$ for the samples with collisions, a

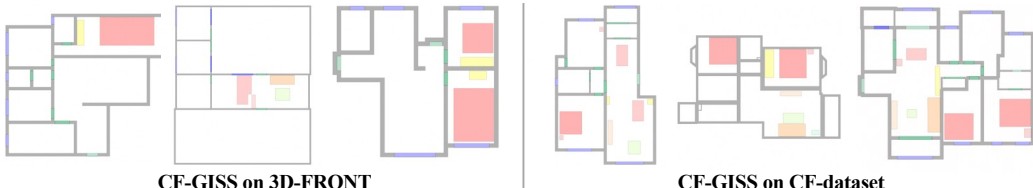

**CF-GISS on 3D-FRONT**                    **CF-GISS on CF-dataset**

Figure 6: Performance of CF-GISS on 3D-FRONT and CF-dataset (ours), where results obtained from training on 3D-FRONT exhibit implausible unfurnished rooms due to artifacts in the original database. Complete results are provided in the Supplementary Materials.

significant 32.22% difference proving the efficacy of CF-GISS in identifying and preventing object collisions as OODs.

## 5.2 EVALUATION ON CF-DATASET

We trained CF-GISS separately with floor plan image and text conditioning on our CF-dataset.

**Dataset and Implementation**    The CF-dataset comprises 9,706 scenes and is ready for use without the need for data cleaning or preprocessing. For both experiments, we use 80% of the dataset for training and the remaining 20% for testing. The GPU infrastructure, model hyperparameters, and training strategies are similar to those detailed in Section 5.1. For image conditioning, we generate the floor plan image for each scene using the color scheme as described in Table 1. For text conditioning, we parse the JSON file of each scene to extract the total area, room count, and categories, which are used to generate the corresponding textual description.

**Results**    *i)* **Image Conditioning** - As seen in Figure 7, CF-GISS is capable of synthesizing diverse and plausible scene layouts and is robust to irregular or slanted room shapes. Similar to the approach in the previous subsection, we also computed metrics including FID, KID, POR, and PIoU. Compared to the metrics obtained on the 3D-FRONT dataset, our CF-dataset yielded relatively higher FID and KID values. This discrepancy could be attributed to the greater variety of object placements and the reduced number of empty rooms in CF-dataset, as illustrated in Figure 6. The collision-related metrics (POR and PIoU) are comparable between the two datasets, likely due to YOLO detection errors (since the original layout images are collision-free) resulting in bounding box collisions. *ii)* **Text Conditioning** - As seen in Figure 8, CF-GISS is capable of synthesizing diverse and plausible scene layouts and floor plans, with approximate areas and specified numbers of living rooms, bedrooms, kitchens, and bathrooms, adhering to the text descriptions. The functional distribution of rooms is coherent, with logical circulation and appropriate furniture arrangement.

## 6 LIMITATIONS, DISCUSSIONS, AND CONCLUSIONS

*i)* **CF-dataset** - While the CF-dataset includes extensive floor plans and scene layout data, the associated 3D models follow a uniform artistic style, lacking diversity in designs and corresponding attributes, which are essential for training text-guided stylistic indoor scene synthesis. The inherent subjective nature of design styles presents a challenge in gathering comprehensive dataset samples, which we leave to future work. *ii)* **CF-GISS** - While the text conditioning of CF-GISS can theoretically extend to various controls, such as spatial layouts (*e.g.*, placing the sofa along the east wall) and stylistic features (*e.g.*, a Bohemian sofa), without significantly altering the pipeline or model architectures, this requires obtaining additional annotated data for such attributes, yet not introducing substantial further novelty in methodology design. Hence, we focus on collision-free scene layout generation, which addresses a prevalent practical limitation in existing approaches, and leave the incorporation of stylistic and spatial controls with texts to future work. Meanwhile, CF-GISS currently does not accommodate natural object overlaps, such as placing a lamp on a table. However, this can be addressed by applying multiple layers of scene layouts, which also effectively prevents vertical collisions, a feature we leave for future work. Another limitation of CF-GISS is its reliance on YOLO (Jocher et al., 2023) for high-accuracy object detection, which struggles to detect the angles of square targets (Ultralytics, 2024). This can lead to bounding box collisions despite the layout images being collision-free. We discuss additional failure cases in Appendix Figure 10.

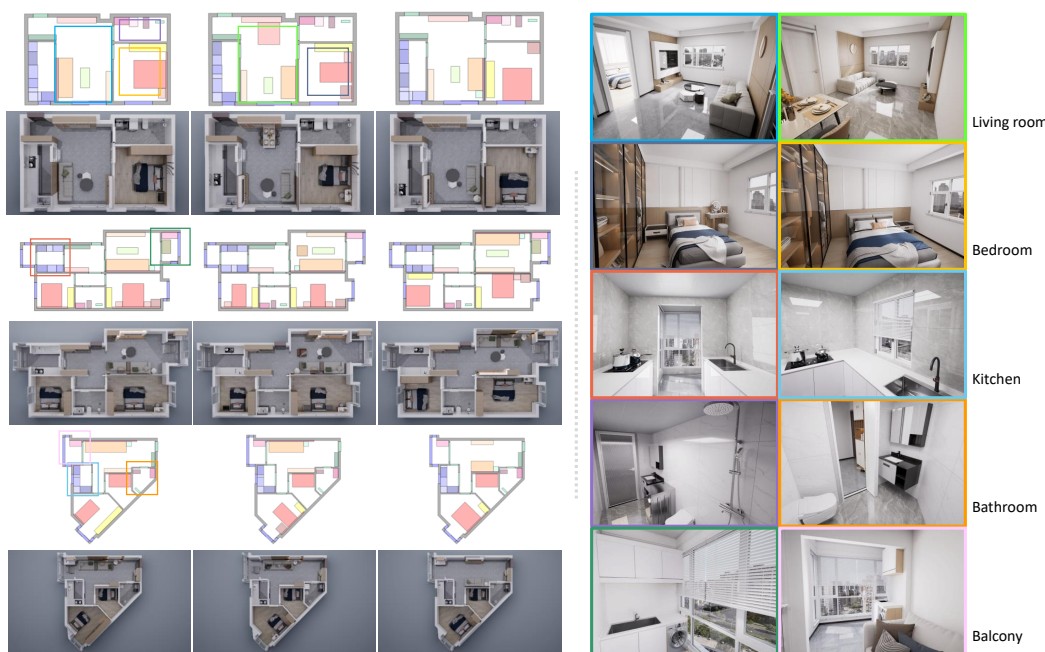

Figure 7: **Left** - Visualization of synthesized layouts by CF-GISS conditioned on three floor plans. We present three diverse layouts synthesized by CF-GISS for each floor plan. Note that CF-GISS is robust to irregular and slanted room shapes. Additional results are provided in Appendix Figure 11, with complete results available in the Supplementary Materials. **Right** - Photorealistic rendering of synthesized scenes by CF-GISS. We present the living rooms and bedrooms with identical camera viewpoints and floor plan structures to demonstrate the diversity of CF-GISS. The correspondence between the layout on the left and the rendering on the right is indicated by matching colored frames.

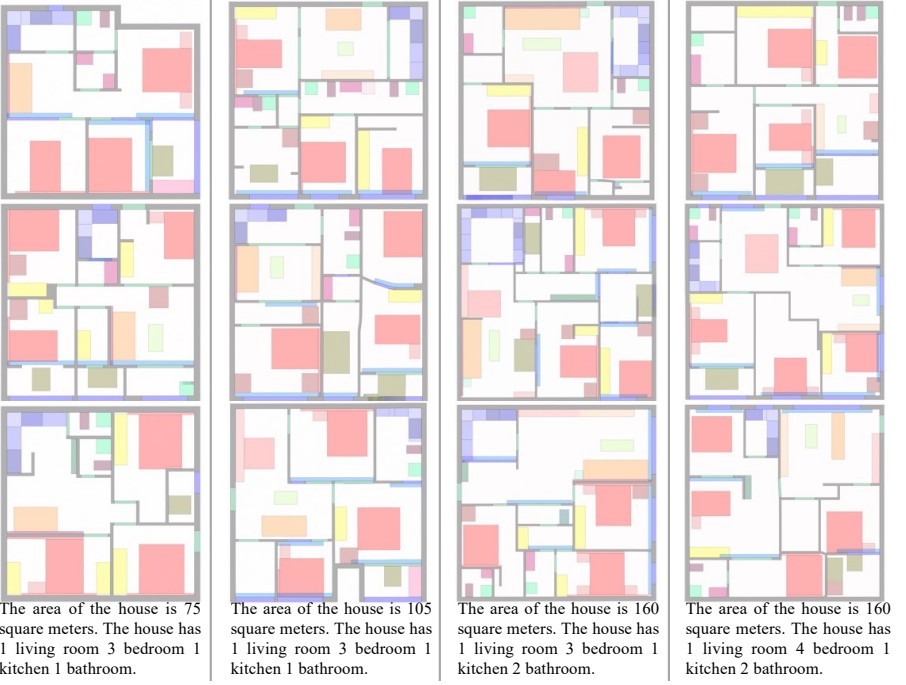

Figure 8: Visualization of text-to-layout generation by CF-GISS trained on our CF-dataset. Floor plans of different room sizes all fill the entire canvas, with the wall thickness set to 24 cm for all scenes. Hence, the room size can be inferred from the thickness of the gray walls, which is consistent with the raw training data.

## 7 ETHICS STATEMENT

In this work, we adhere to responsible research practices, ensuring that the datasets and methods used comply with legal and ethical standards. The dataset created and utilized in this study was sourced and generated without infringing on the rights of individuals or organizations. We have ensured that the data is free of personally identifiable information, and that no harm has been inflicted on any subjects during the research process. Furthermore, the research addresses technical challenges in generative indoor scene synthesis, with no direct societal or environmental risks.

## 8 REPRODUCIBILITY STATEMENT

To ensure the reproducibility of our work, we have provided the source code in the supplementary materials, along with detailed descriptions of our methodology and experimental setup in the main paper. The CF-GISS model, including hyperparameters, training procedures, and evaluation metrics, has been thoroughly documented. Upon acceptance, we will publicly release the source code, pretrained models, and our novel CF-dataset to facilitate verification and further research in this field.

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

# A APPENDIX

## A.1 CF-GISS INFERENCE EFFICIENCY

On a single RTX 8000 GPU, the diffusion model inference takes approximately 18 seconds; YOLO object detection takes around 40 milliseconds; 3D model matching and scene construction take about 100 milliseconds; and rendering, performed using the UE engine, takes approximately 30 seconds for a 2K image and around 120 seconds for a 4K image. While rendering is the most time-consuming module, it is an independent component that can be flexibly replaced with any real-time rasterization-based renderer when efficiency is a priority. We chose a ray-tracing-based renderer for photorealistic quality.

## A.2 ABLATION STUDIES

We additionally conducted an ablation experiment where the diffusion model generated 2D layouts as segmentation maps, i.e., quantized grayscale images with precision consistent with the number of household item categories, instead of RGB images. The diffusion model shares the same architecture as described in Section 3.1. We present qualitative results in Figure 9, which indicate that the diffusion model fails to capture meaningful 2D layouts when directly generating segmentation maps. In Figure 9, for better visualization and comparison, we additionally converted each segmentation map into an RGB-based image, consistent with the color scheme used in the main experiments. Meanwhile, we conducted a quantitative evaluation, which yielded an FID of 13.34 and a KID of 0.014 between the generated segmentation maps and the ground truth, as well as an FID of 328.20 and a KID of 0.52 between the RGB layouts converted from the generated segmentation maps and the ground truth. All the metrics are consistently worse compared to CF-GISS as reported in Table 2 (Entire House). We believe this is due to the insufficient precision of the segmentation maps, which seriously compromises the robustness of the diffusion model. Compared to full-range RGB images, minor inaccuracies in generated segmentation maps can lead to significant errors. This further validates our current choice of the CF-GISS pipeline, where the use of RGB-based layouts with 255 precision, combined with a YOLO detection step, improves the robustness of our approach.

Listing 1: Example JSON data format

```json
{
    "rooms": [
        {
            "roomId": "D5F19A0446724E5EBE4AF38251000000",
            "roomName": "living", # inner room
            "roomType": 1,
            # 2d coords x, y in cm, at least 3 points, polygon
            "wallPoints": [[171.65, 241.5], [651.66, 241.5], ...]
        },
        {
            "roomId": "D5F19A0446724E5EBE4AF38251000114",
            "roomName": "out_room", # outter room
            "roomType": 0,
            "wallPoints": [[171.65, 241.5], [651.66, 241.5], ...] # 2d
                coords
        },
    ],
    "windowsDoors": [ # 3d bounding box data
        {
            "type": "door",
            # box center position x,y; height to floor z
            "pos": [717.32, 737.0, 0],
            "length": 95,
            "width": 12,
            "height": 210,
            "rotate": 100 # roate angle in degree
        },
        {
```

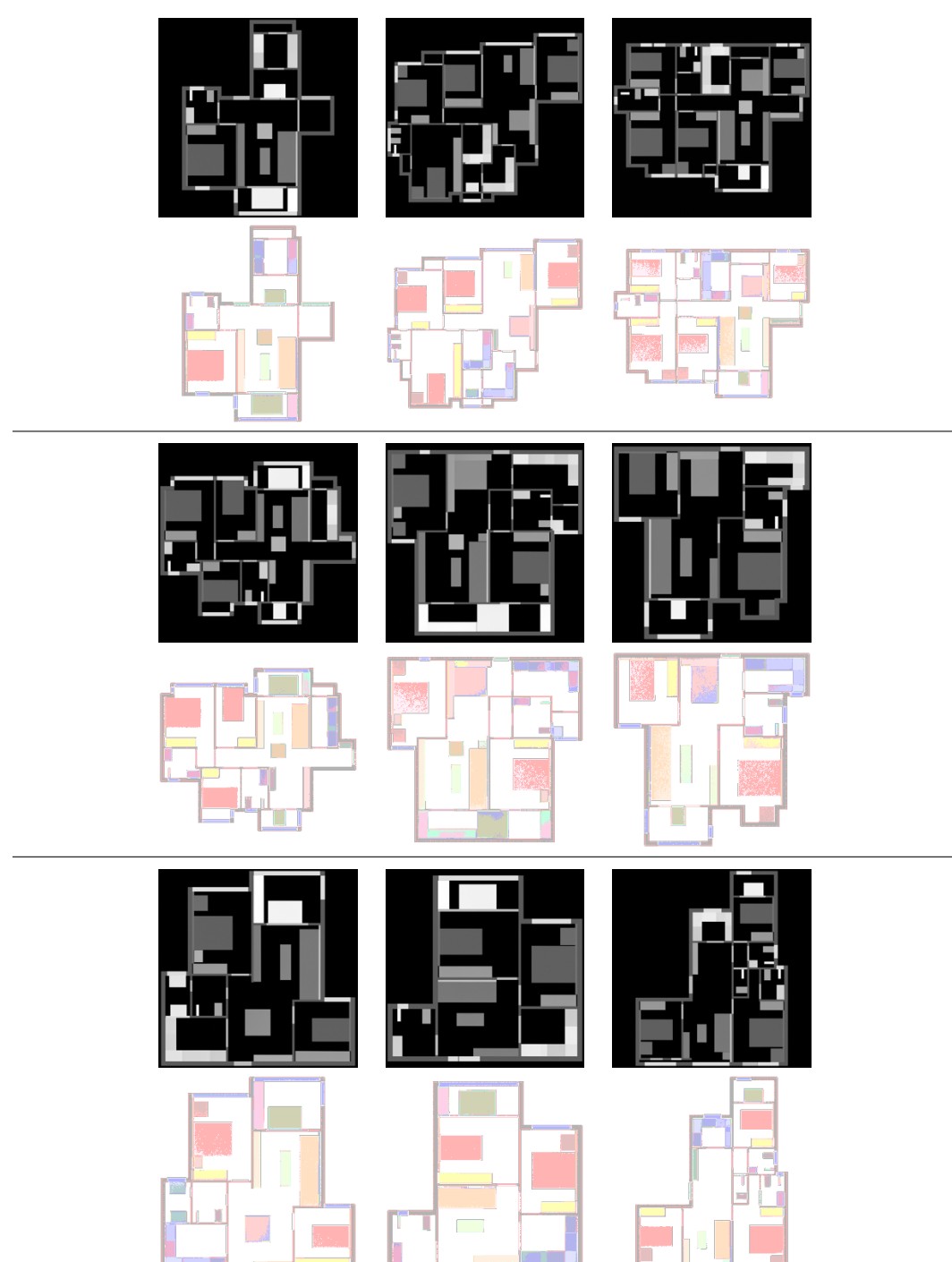

Figure 9: Visualization of ablation study results. Grayscale images represent segmentation maps directly generated by the diffusion model. Each segmentation map is converted into an RGB-based layout image, consistent with the color scheme used in the main experiments. While inaccuracies in the grayscale images are not directly observed due to insufficient precision, the converted color images clearly reveal that the generated segmentation maps contain significantly more artifacts compared to the layouts generated by CF-GISS.

```
        "type": "window",
        "pos": [657.66, 945.12, 90],
```

```
810          "length": 153.75,
811          "width": 12,
812          "height": 110,
813          "rotate": 90.0
814      }
815    ],
       "furniture": [ # 3d bounding box data, same with windows and doors
816        {
817          "type": "coffee_table",
818          "pos": [569.91, 1844.75, 0],
819          "length": 76.0,
820          "width": 94.0,
821          "height": 99,
          "rotate": 180.0
822        },
823        {
824          "type": "sofa",
825          "pos": [411.66, 169.45, 0],
826          "length": 185.3,
827          "width": 120.1,
828          "height": 99,
          "rotate": 0.0
829        }
830    ]
    }
831
832
833
```

Listing 2: Furniture map from 3D-FRONT to Ours

```
834  {
835    "Nightstand": "bedside table",
836    "Wardrobe": "cabinet",
837    "Three-Seat / Multi-seat Sofa": "sofa",
838    "Dining Table": "dining table",
839    "Coffee Table": "coffee table",
840    "Loveseat Sofa": "sofa",
841    "Children Cabinet": "cabinet",
842    "Drawer Chest / Corner cabinet": "cabinet",
843    "King-size Bed": "bed",
844    "TV Stand": "tv cabinet",
845    "Sideboard / Side Cabinet / Console": "dining cabinet",
846    "Lazy Sofa": "leisure_sofa",
847    "Dressing Table": "table",
848    "Wine Cabinet": "dining cabinet",
849    "L-shaped Sofa": "sofa",
850    "Corner/Side Table": "side coffee table",
851    "Bookcase / jewelry Armoire": "cabinet",
852    "Kids Bed": "bed",
853    "Sideboard / Side Cabinet / Console Table": "table",
854    "Bed Frame": "bed",
855    "Shoe Cabinet": "shoe cabinet",
856    "Three-Seat / Multi-person sofa": "sofa",
857    "Double Bed": "bed",
858    "Bunk Bed": "bed",
859    "Desk": "table",
860    "Two-seat Sofa": "sofa",
861    "Tea Table": "coffee table",
       "Couch Bed": "bed",
       "Single bed": "bed",
       "Chaise Longue Sofa": "sofa",
       "U-shaped Sofa": "sofa"
862  }
863
```

| Room | Furniture | 3D-FRONT | CF-dataset (ours) |
|---|---|---|---|
| Bedroom | Bed | 10620 | 24354 |
| | Cabinet | 17649 | 19365 |
| | Bed Background | 0 | 16619 |
| | Bedside Table | 14333 | 10439 |
| | Table | 8318 | 9359 |
| Living Room | Leisure Sofa | 237 | 8953 |
| | Sofa | 6564 | 8430 |
| | TV Cabinet | 6821 | 7935 |
| | Sofa Background | 0 | 7019 |
| | Coffee Table | 6565 | 7005 |
| | Dining Cabinet | 1169 | 5368 |
| | Shoe Cabinet | 0 | 4817 |
| | Single Sofa | 0 | 3939 |
| | Dining Table | 5822 | 3444 |
| | Side Coffee Table | 6300 | 4195 |
| Kitchen | Single Door Cabinet | 0 | 15889 |
| | Double Door Cabinet | 0 | 14156 |
| | Cooker Cabinet | 0 | 6904 |
| | Sink Cabinet | 0 | 6773 |
| | Electrical Cabinet | 0 | 2081 |
| | Refrigerator | 0 | 1307 |
| Bathroom | Shower | 0 | 15174 |
| | Toilet | 0 | 15026 |
| | Washbasin | 0 | 12517 |
| | Washing Machine | 0 | 970 |
| Balcony | Washing Machine Cabinet | 0 | 4153 |

Table 3: Comparison of furniture occurrences between 3D-FRONT and CF-dataset.

| | Empty Room Rate | POR | PIoU |
|---|---|---|---|
| 3D-FRONT | 0.5906 | 0.0361 | 0.2547 |
| CF-dataset (ours) | 0.2902 | 0.0044 | 0.0018 |

Table 4: Comparison of data quality statistics between 3D-FRONT and CF-dataset.

| | Living | Bedroom | Kitchen | Bathroom | Balcony | Empty Room Rate |
|---|---|---|---|---|---|---|
| 3D-FRONT | 1813 | 4041 | 0 | 0 | 0 | 0.5906 |
| CF-dataset (ours) | 15115 | 40983 | 8262 | 16351 | 8262 | 0.2902 |

Table 5: Comparison of non-empty room statistics between 3D-FRONT and CF-dataset.

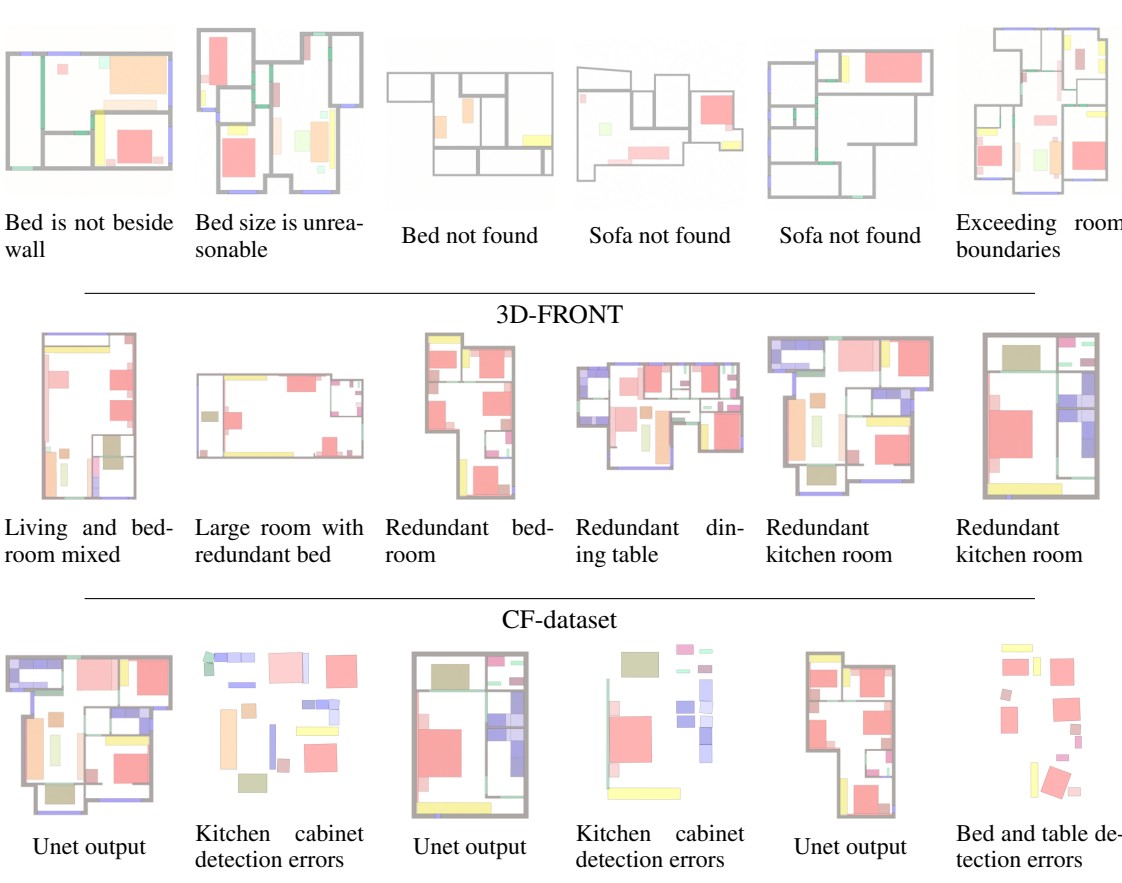

Bed is not beside wall | Bed size is unreasonable | Bed not found | Sofa not found | Sofa not found | Exceeding room boundaries

3D-FRONT

Living and bedroom mixed | Large room with redundant bed | Redundant bedroom | Redundant dining table | Redundant kitchen room | Redundant kitchen room

CF-dataset

Unet output | Kitchen cabinet detection errors | Unet output | Kitchen cabinet detection errors | Unet output | Bed and table detection errors

YOLO detection errors

Figure 10: Failure cases of CF-GISS. **Top** - Failure cases of CF-GISS on the 3D-FRONT dataset such as missing or unreasonably placed furniture are due to the inherent erroneous data samples found in 3D-FRONT. **Middle** - CF-GISS occasionally misidentifies incorrect room categories, and thus misplacing household items. We anticipate this issue to be mitigated by providing a larger amount of training data. However, note that this is a limitation unique to our method that is capable of floor-plan-conditioned layout generation. Training CF-GISS solely on a single category of room similar to Tang et al. (2024), Lin & Mu (2024), and Yang et al. (2024) will not encounter this issue. **Bottom** - CF-GISS heavily relies on the accuracy of YOLO detection, which also occasionally fails, as discussed in Section 6.

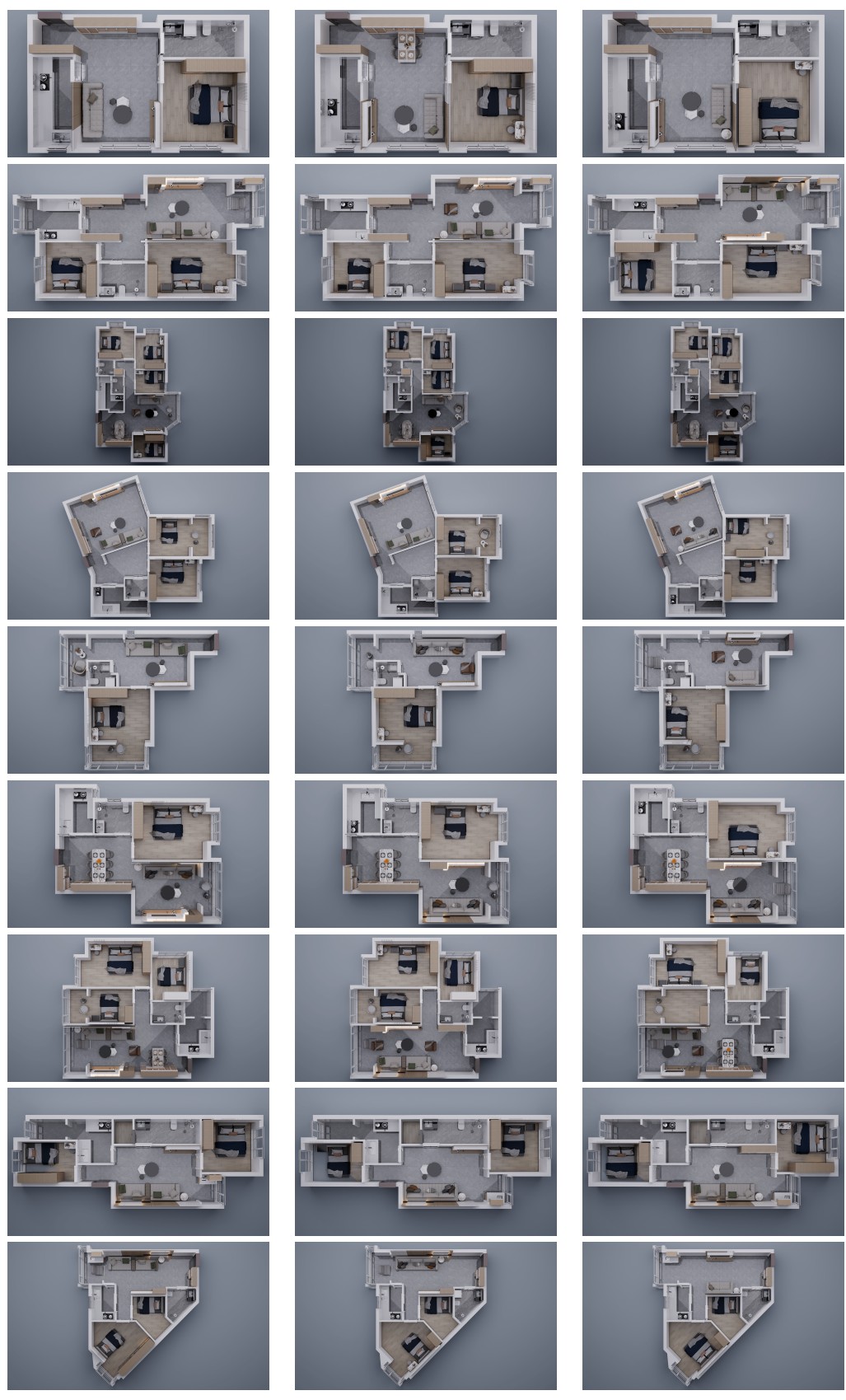

Figure 11: Additional photorealistic rendering of diverse synthesized layouts by CF-GISS conditioned on floor plans.

