# OpenReview forum: "CF-GISS: Collision-Free Generative 3D Indoor Scene Synthesis with Controllable Floor Plans and Optimized Layouts"
_ICLR.cc/2025/Conference — Submitted to ICLR 2025_

### Official Review · Reviewer_Vvb4 · 2024-10-31

**Soundness:** 2
**Presentation:** 3
**Contribution:** 2
**Rating:** 3
**Confidence:** 4

**Summary:**

This paper presents CF-GISS, a novel framework for generative 3D indoor scene synthesis that guarantees collision-free scene layouts by leveraging an image-based intermediate layout representation. By treating collision artifacts as out-of-distribution (OOD) scenarios during generation, CF-GISS effectively prevents such artifacts. The framework takes floor plans or textual descriptions as input conditions to generate a coherent and realistic room layout. Moreover, this paper introduces a new indoor scene dataset that offers expanded coverage of household items and room configurations.

**Strengths:**

1. Novel approach. The use of an intermediate RGB image representation for scene layout is a clever way to leverage image processing tools for collision detection and avoidance.
2. Good performance. The paper claims and demonstrates significant improvements over existing methods on the 3D-Front dataset, particularly in the elimination of object collisions, which is a notable achievement.
3. Dataset Contribution. The introduction of a new, larger dataset with improved quality and expanded coverage is a substantial contribution to the field, providing a valuable resource for future research.
4. Clear presentation. The paper presents the methodology in a clear and coherent manner, consistently emphasizing the significant contribution of collision-free scene generation.

**Weaknesses:**

1. The proposed image-based intermediate layout representation lacks a theoretical guarantee of zero collision, as the authors fail to provide rigorous proof to support their claim.
2. Misaligned results. The renderings presented in Figure 2, depicting living rooms, bedrooms, and other scenes, exhibit significant misalignment with the corresponding floor plans above. Furthermore, the furniture within these renderings is also misaligned with the retrieved objects below. Similarly, the renderings in Figure 7 lack corresponding floor plans, which raises concerns about the authenticity of the results.
3. Incomplete comparison. Previous work PhyScene [1] also explored the collision problem while generating the indoor scenes.
4. Invalid room layout. The room layouts presented in Figure 8 contain enclosed areas, which raises questions about the feasibility of entering these rooms without doors.
5. Duplicate furniture. The authors claim that their new dataset, CF-dataset, comprises diverse furniture with improved quality. However, upon closer inspection, repetitive furniture (such as beds and coffee tables) is evident in Figures 2, 7, and 10, which contradicts their assertion.
6. As described in Section 3.2, some object properties are obtained based on predefined standards, which may compromise the diversity of scene generation.

[1] PhyScene: Physically Interactable 3D Scene Synthesis for Embodied AI, Yang. et al, CVPR 2024

**Questions:**

In Line 244, the paper mentions that wall and floor materials are procedurally sampled. Could the authors provide a more detailed explanation of this process? How does this method ensure style and aesthetic coherence within the generated indoor scenes?

---

> ### Author Response · Authors · 2024-11-21
>
> Thank you for your constructive feedback.
>
> ## Theoretical Analysis
> While we acknowledge that we did not provide rigorous mathematical proof for zero collisions, our manuscript includes a theoretical analysis (lines 422–444 in Section 5.1) from the perspective of out-of-distribution (OOD) samples, which explains why CF-GISS performs so effectively.
>
> In diffusion models, the training loss is computed as the reconstruction error of the data given the noise, serving as an approximation of the negative log-likelihood (NLL). If the model has been adequately trained on the data distribution, a high training loss for a sample indicates a high NLL, suggesting an OOD scenario that is unlikely to occur during sampling. Our analysis shows that samples with collisions exhibit a training loss significantly higher than collision-free samples. Consequently, samples with collisions can be considered OOD samples that CF-GISS is highly unlikely to generate.
>
> Reviewer CTZu also highlighted that:
> > “As we have already trained diffusion models to understand so much visual information, current techniques surely perform well in determining whether objects are overlapping in the scene. Plenty of experiments also prove this.”
>
> We are happy to soften our claim if any part of our original text suggests that our method is entirely free of unnatural collisions. Most importantly, our empirical experiments provide strong evidence of our method’s substantial (rather than marginal) improvements over existing approaches for collision avoidance. These existing methods are largely unsuitable for real-world applications due to the pervasive presence of collision artifacts. We believe this is a shared concern that warrants attention and resolution, and that our work makes significant progress toward mitigating collision artifacts.
>
> ---
>
> ## Figures and Concern About Result Authenticity
> We apologize for the oversight. Figure 2 primarily serves as an illustrative diagram to explain our methodology rather than to present results. We have, however, corrected this diagram in the revised manuscript to also present accurate renders.
>
> Regarding Figure 7, it is impractical to present renderings of all rooms and all layouts due to the limited space available. We have introduced matching color frames to indicate the correspondence of renderings and layouts in the revised manuscript.
>
> We take the authenticity of our results very seriously. Despite the apparent ambiguities, all the presented results are 100% authentic, and we have been fully transparent about CF-GISS’s limitations (Section 6). Additionally, our initial manuscript includes a reproducibility statement on page 11. Our source code, data, and complete results are fully available in the supplementary materials and will be publicly released upon acceptance.
>
> ---
>
> ## Comparison with PhyScene
> Thank you for your suggestion. We have conducted additional experiments to compare our results with PhyScene and have included the new results in the revised manuscript (Section 5.1, Figure 5, Table 2). To ensure PhyScene’s best performance, we used the checkpoint provided by its authors, which was also trained on 3D-FRONT. Both qualitative and quantitative results demonstrate that CF-GISS significantly outperforms PhyScene.
>
> We would also like to highlight that PhyScene avoids furniture collisions and prevents furniture from exceeding room boundaries by incorporating IoU-related loss functions. This approach imposes only a soft constraint, which differs fundamentally from our method. Additionally, PhyScene’s floor plan constraints do not account for doors and windows, which could lead to furniture blocking these features during generation. Finally, CF-GISS is the only approach that has demonstrated house-wide layout generation rather than being limited to single-room layouts. PhyScene did not offer empirical results in this regard.

---

> ### Author Response · Authors · 2024-11-21
> **Continued**
>
> ## Invalid Room Layout
> We acknowledge that CF-GISS occasionally generates failure cases, such as enclosed areas when using text conditioning (this issue does not occur with floor plan image conditioning). We believe this is a common artifact observed in diffusion models—purely data-driven models—when trained on insufficient data samples. For instance, text-to-image models are well known to generate anomalies such as humans with six fingers. Addressing such limitations is one of the key motivations behind the introduction of our new CF-dataset.
>
> We have also trained CF-GISS on a private dataset of over 100,000 indoor scenes (CF-dataset currently consists of 9,706 scenes), which we invested significant time and resources to collect and organize. Due to privacy constraints, we are unable to release the entire dataset but can provide the CF-dataset at its current scale. Training CF-GISS on the full dataset of 100,000+ scenes has significantly reduced artifacts, such as invalid room layouts. While we cannot release the complete dataset, we are happy to make the model checkpoint trained on this larger dataset freely available upon acceptance.
>
> We would also like to highlight that CF-GISS is the only approach that has demonstrated the ability to generate house-wide layouts conditioned on text-guided floorplans. DiffuScene, InstructScene, and PhyScene have only demonstrated the generation of layouts for individual rooms. While our solution is not perfect, we believe it will substantially benefit the community. Additionally, we have proposed promising solutions (e.g., larger datasets) to further improve our performance.
>
> ---
>
> ## Duplicate Furniture
> We would like to clarify that CF-dataset is a 3D scene layout dataset that introduces additional categories of household items for scene layouts compared to 3D-FRONT, rather than adding variations to the styles of individual household item categories. For instance, 3D-FRONT lacks several household items in its scenes, such as toilets and sink cabinets, as shown in Figure 3. However, the CF-dataset does not include, for example, 1,000 variations of toilets. Our initial manuscript also explained this in lines 262-264.
>
> At the object retrieval stage, once an item’s category is established, users can freely associate it with any large public 3D asset datasets, such as Objaverse, for stylistic variations. Similarly, 3D-FRONT has been associated with 3D-FUTURE for this purpose.
>
> While it is possible to link the CF-dataset to such a dataset to incorporate stylistic variations when training CF-GISS, we believe this digresses from the main focus of this paper, which is to generate collision-free, house-wide room layouts. As a result, our initial manuscript has discussed style control as future work in Section 6 (lines 475-480).
>
> ---
>
> ## Predefined Standards
> As discussed above, variation in object styles is not the main focus of this paper. Our manuscript discussed this as future work in Section 6 (lines 475-480).
>
> ---
>
> ## Wall and Floor Materials
> We have preconfigured multiple sets of material style templates from an aesthetic perspective, including styles such as modern, light luxury, and vintage. These templates include a variety of items, such as beds and sofas of different sizes, flooring, wall paint, lights, decorations, and cabinets.
>
> After generating the positions and sizes of major furniture pieces using CF-GISS, we match the furniture to the most suitable items in the template based on the room type and dimensions. Simultaneously, we match appropriate flooring, wall paint, and lights from predefined material templates to the room type. For instance, bedrooms are matched with wooden flooring and wall paint, while bathrooms and kitchens are matched with tiles. This ensures not only a consistent furniture style but also alignment between the furniture and the flooring, wall paint, and lighting styles.
>
> This method is not a core part of the CF-GISS pipeline in terms of novelty, so we only briefly mentioned it in the paper. However, it is essential in the final presentation of results. In the paper, we have rendered all the scenes using the modern style template, as we leave style control to future work.

---

### Official Review · Reviewer_CTZu · 2024-11-02

**Soundness:** 3
**Presentation:** 3
**Contribution:** 3
**Rating:** 6
**Confidence:** 4

**Summary:**

CF-GISS present a novel scene synthesis idea: (1) generate the top-down view of the indoor scene where the objects are shaded in the color related to their semantic label, (2) retrieve specific model for each object and fill them into the place. To train the image generative model in (1), the author creates a nicer scene dataset with more scenes and better object layout. The scene synthesis method addresses the object intersecting problem, which is more visual in the scene’s top-down view comparing to the numerical recording of each bounding box.

**Strengths:**

The idea of solving “object intersection” through the top-down image is elegant and sound. As we have already trained diffusion models for understanding so much visual information, current techniques surely perform well in telling whether there are objects overlapping in the scene. Plenty of experiments also proves this. Furthermore, the method won’t be restricted with the “maximum object in the scene” and larger scenes (such as the entire house).can be generated for once. Providing a better scene dataset also brings more convenience to other researcher.

**Weaknesses:**

Let alone the accuracy of YOLO and the result of image generation training, I still have a trivial worry about whether addressing the generating issue with image may break the inner structure of object. Because when rendering the top-down view and generating these top-down views from the diffusion model, the objects are all presented in pixels. What would the pixels doesn’t form objects?
Furthermore, the valid intersections through the top-down view can never be presented with the method, e.g. the pendant lamp above other furniture, or a vase on a nightstand, or some chairs are put under the table. Would that strongly effect the synthetic results?

**Questions:**

No

---

> ### Author Response · Authors · 2024-11-21
>
> Thank you for your constructive feedback.
>
> We agree that top-down view layouts do not preserve the inner structures of objects, such as their styles and shape details. Previous work, such as DiffuScene, has introduced “shape vectors” for each object to represent “inner structures”. This is one of the reasons we use the 2D layout only as an intermediate representation, with the scene graph serving as the final layout representation. In the future, we could potentially generate similar shape vectors for each node in the scene graph, conditioned on images, user text specifications, or other inputs.
>
> We acknowledge that there could be reasonable furniture overlaps, as we previously discussed in Section 6:
>
> > "CF-GISS currently does not accommodate natural object overlaps, such as placing a lamp on a table. However, this can be addressed by applying multiple layers of 2D layouts, which also effectively prevents vertical collisions, a feature we leave for future work."
>
> For example, we can assume that the second layer is always positioned on top of the first layer and potentially use the first layer as conditioning for generating the second. We believe this approach would effectively address scenarios involving natural object overlaps, such as a pendant lamp hanging above other furniture or a vase placed on a nightstand.
>
> For the specific scenario where parts of chairs are positioned under a table, a simpler solution would be to treat the table and chairs as a single 3D model, as dining tables are typically paired with chairs for consistent styles and sizes.
>
> While we leave this feature for future work, we would like to highlight that most existing approaches completely fail to avoid unnatural collisions, as shown in Figure 5, making these methods utterly unsuitable for real-world applications. Addressing this issue is therefore the main focus of this paper.

---

### Official Review · Reviewer_wFS6 · 2024-11-03

**Soundness:** 3
**Presentation:** 3
**Contribution:** 3
**Rating:** 8
**Confidence:** 4

**Summary:**

This paper introduced a new framework, CF-GISS, for creating generative 3D indoor scenes that prevents collisions in scene layouts by using an image-based intermediate representation. This method mitigates collisions during generation, even in out-of-distribution scenarios. It allows for layout generation based on floor plans that can be controlled via images or text, and gives coherent layouts that adapt to different geometric and semantic structures.

The framework achieves state-of-the-art results on the 3D-FRONT dataset, and gives high-quality, collision-free scenes, accommodating various floor plan designs. Besides, it introduces a novel dataset with enhanced coverage of household items and room configurations, along with improved data quality.

The paper is well-written and easy to follow. I enjoy reading it.

**Strengths:**

I think the major contribution of this paper is the new framework that supports multimodal conditions - it synthesizes collision-free generative 3D indoor scenes with customizable floor plans based on images or text prompts, optimized room configurations, and photorealistic rendering.

To support training, they synthesized a new dataset with floor plans and scene layouts.

The performance on 3D-FRONT achieves the state-of-the-art.

**Weaknesses:**

The weakness of this paper is the relative long and non-differentiable pipeline. e.g., for the object retrieval module, it means the pipeline should come with a large object dataset to finish the whole process of scene synthesis. However, I know that most current scene synthesis works have an object retrieval phase.

Another suggestion is - Since existing scene datasets are much smaller than concurrent 3D object datasets for generative tasks, it is easy to overfit during synthesis. It would be more informative to compare the synthesized scenes with their nearest neighbours in the training set, to compare and evaluate if the methods overfit, and to evaluate if the scale of the datasets is enough.

**Questions:**

How efficient of this current model in inference? I noticed that there are many off-the-shelf modules, e.g., object detection, retrieval and rendering. It is quite a long pipeline.

---

> ### Author Response · Authors · 2024-11-21
>
> Thank you for your constructive feedback.
>
> ## Model Performance Associated with Dataset Scales
> We acknowledge that our pipeline relies on multiple sequential and non-differentiable modules, which necessitate several large datasets for achieving reasonable performance, similar to most procedural scene synthesis works. This motivated the introduction of our new CF-dataset.
>
> In our early experiments, we used the 3D-FRONT dataset. However, we found it was highly prone to overfitting on smaller datasets, as the 3D-FRONT dataset contained only 4,847 usable samples after data cleaning. Recognizing these data quality limitations, we invested significant time and resources in collecting and organizing a private dataset of over 100,000 indoor scenes, which became the foundation of our subsequent research. Due to privacy policies, we cannot fully disclose this dataset. Instead, we will make a subset of 9,706 scenes (CF-dataset) publicly available, which is ready to use without requiring additional cleaning.
>
> Our algorithm performs significantly better on this 9,706-sample dataset compared to the limited 3D-FRONT dataset, as reported in Section 5.2 and Figure 6. Additionally, our model trained on the full 100,000+ scenes achieves even better results, with substantially fewer failure cases. While we are unable to release the full 100,000+ scenes publicly, we are happy to provide the checkpoint of our model trained on this larger dataset and will include a discussion of this in the final version of the paper.
>
> Regarding overfitting, we are pleased to report that our model is well-fitted after training on the current CF-dataset (consisting of 9,706 scenes), which will be made publicly available upon acceptance. As shown in Figure 7, CF-GISS is capable of generating diverse 2D layouts conditioned on the same floorplan, despite the original CF-dataset containing only a single scene layout per floorplan. We believe this demonstrates strong evidence that the model does NOT overfit.
>
> ## Inference Efficiency
> On a single RTX 8000 GPU:
> - The diffusion model inference takes approximately **18 seconds** (although this can be potentially reduced with advanced diffusion solvers).
> - YOLO object detection takes around **40 milliseconds**.
> - 3D model matching and scene construction take about **100 milliseconds**.
> - Rendering, performed using the UE engine, takes approximately **30 seconds** for a 2K image and around **120 seconds** for a 4K image.
>
> While rendering is the most time-consuming module, it is an independent component that can be flexibly replaced with any real-time rasterization-based renderer when efficiency is a priority. We chose a ray-tracing-based renderer for photorealistic quality. It is also worth noting that relying on off-the-shelf renderers is a common feature across procedural 3D scene synthesis methods, including InstructScene and DiffuScene.
>
> We have included this report in the revised manuscript (Appendix A.1).

---

### Official Review · Reviewer_iEvZ · 2024-11-04

**Soundness:** 2
**Presentation:** 2
**Contribution:** 3
**Rating:** 3
**Confidence:** 4

**Summary:**

This paper introduces a 3D indoor scene layout generation method based on 2D diffusion models. Instead of directly denoising parameterized objects and location features, this method uses 2D floor plan images as an intermediate representation to train a 2D diffusion model. Then, the authors use an object detection model to extract scene graphs from the generated layout image and reconstruct the 3D scene from it. The authors also constructed a new indoor scene dataset which is 1.4 times larger than 3D-FRONT dataset.

**Strengths:**

1. The authors constructed a new dataset which is 1.4 times larger than 3D-FRONT, with expanded household item categories with 26 super-categories and includes more scenarios like kitchen, bathrooms, and balconies. They also reduced several problems like empty scenes, object exceeding boundaries and overlapping compared to 3D-FRONT.

2. Incorporating the 2D layout image and 2D diffusion model during the generation enhances the model capacity of understanding the spacial relationships of each object and the floor plan.

**Weaknesses:**

1. The overall writing can be improved, some expressions are not clear enough.

2. I am quite confused that the authors append an extra object detector to detect objects from the generated layout images. Why not directly generate a segmentation map since they are basically trained from a segmentation map? Why not directly train on a quantized segmentation mask? What is the purpose of an additional object detector as it may introduce additional noise to the process? And it may further improve the ability to reduce collisions.

2. Is the scene graph extraction really necessary when you already have a 2D layout map? Scene graphs are necessary in those 1D diffusion-based works since there are no explicit representations of the spatial relationships, but what does scene graphs contributes when you already have an explicit 2D layout map?

3. The captions of Figure 3 are confusing. It seems that the authors were trying to explain the existing problems in the 3D-FRONT dataset, but the captions given are "Qualitative comparison of CF-dataset (ours) with 3D-FRONT." However, there is no comparison between the two datasets, just some demonstration of erroneous cases.

4. In table 2, the authors compared DiffuScene and InstructScene with the proposed method only on 3D-FRONT but not on the proposed CF-dataset, as I checked, both these method have released their codes, so why not compare the results on the proposed CF-dataset？

5. The authors proposed two conditions, image floor plan or text descriptions for the generation procedure as alternatives. However, these two conditions are not exclusive， it is more reasonable to use both the conditions together when you have a floor plan and use text prompts to describe the furnitures in the rooms.

5. No ablation studies are included in the experiments.

**Questions:**

1. Sometimes collision/occlusions in the vertical top views may not indicate wrong layouts, like part of the chairs can be put under the table, is it really necessary to exclude all the collision/occlusion during the generation?

---

> ### Author Response · Authors · 2024-11-21
>
> Thank you for your constructive feedback.
>
> ## Unclear Expressions
> We apologize for any confusion in our writing. We would be happy to clarify any details if you could specify which expressions require further explanation.
>
> ## Necessity of Object Detector and Direct Generation of Segmentation Maps
> The YOLO detection model provides the bounding boxes, object categories, positions, and orientations of each household item, all of which are essential for accurate object retrieval and placement.
>
> While it is possible to perform graph extraction and object retrieval based on the segmentation map, additional postprocessing would still be required to derive the bounding boxes, positions, sizes, and orientations for each object—a process prone to errors. Additionally, since the scene may contain **multiple instances** of the same object type (e.g., beds), generated segmentation maps cannot be directly utilized without a subsequent detection module to extract individual objects.
>
> We chose the current pipeline because YOLO object detection is a well-established framework for RGB images. Our experiments have shown that YOLO detection can efficiently and accurately perform the targeted tasks, without the need for postprocessing, while also providing robustness against various irregularities. These irregularities include cases such as furniture color blocks not perfectly aligned with associated categories or edges exhibiting jagged patterns, which are common in diffusion-generated images. Directly using diffusion-generated results as ground-truth segmentation maps without robust postprocessing is likely to exacerbate artifacts. Thus, we believe the YOLO detection module is an essential component of our pipeline. While YOLO detection is not entirely error-free, our quantitative results demonstrate substantial (rather than marginal) improvements over existing methods in avoiding collisions. Failure cases due to inaccurate YOLO detection are extremely rare.
>
> Nonetheless, we conducted additional experiments where the diffusion model directly generated segmentation maps, i.e., quantized grayscale images with precision consistent with the number of household item categories. We have included and discussed both qualitative and quantitative results in Appendix A.2 of the revised manuscript. The results indicate that diffusion fails to capture meaningful 2D layouts without the object detection module when directly generating segmentation maps. We believe this is due to the insufficient precision of the segmentation maps, which seriously compromises the robustness of the diffusion model. Compared to full-range RGB images, minor inaccuracies in generated segmentation maps can lead to significant errors. This further validates our current choice of the CF-GISS pipeline, where the use of RGB-based layouts with 255 precision, combined with a YOLO detection step, improves the robustness of our approach.
>
> ## Scene Graph
> We extract a structured graph representation from the 2D layout for two purposes:
> 1. We apply a three-level graph structure that stores the affiliation relationships between household items and their associated rooms. These relationships, which are not explicitly available in the layout, require postprocessing steps, such as room detection, to identify and store them separately.
> 2. We use this approach to demonstrate that our pipeline can be seamlessly integrated into the majority of existing methods that use graph representations for indoor scene generation. This representation allows additional information to be stored in the nodes and edges, enabling further downstream applications.
>
> We advocate that all such graph-based methods should adopt an intermediate 2D layout to avoid collisions.
>
> ## Figure 3 Caption
> Thank you for your comments. Our aim is indeed to demonstrate the erroneous scenes in 3D-FRONT that do not occur in our dataset. To avoid confusion, we have removed “Qualitative Comparison” from the caption.
>
> ## Evaluation of DiffuScene and InstructScene on CF-dataset
> While it is possible to further evaluate DiffuScene and InstructScene on the CF-dataset, our current experiments provide sufficient evidence of our method’s substantial (not marginal) improvements over existing approaches for collision avoidance, along with a theoretical analysis from the perspective of out-of-distribution (OOD) sampling. Most importantly, all prior approaches (including DiffuScene and InstructScene) were solely evaluated on the 3D-FRONT dataset, and thus we have followed this standard practice. The CF-dataset represents an important but independent contribution of our paper, which we believe will significantly benefit the community. Our experiments on the proposed CF-dataset primarily aim to demonstrate its unique strengths compared to the problematic 3D-FRONT dataset.

---

> ### Author Response · Authors · 2024-11-21
> **Continued**
>
> ## Combined Conditioning
> We agree that the two modalities can be combined for better control over generated furniture layouts. However, our current text conditioning is restricted to specifying the floor plan rather than the arrangement of furniture within the rooms. Our manuscript discussed this in Section 6:
>
> > "While the text conditioning of CF-GISS can theoretically extend to various controls, such as spatial layouts (e.g., placing the sofa along the east wall) and stylistic features (e.g., a Bohemian sofa), without significantly altering the pipeline or model architectures, this would require additional annotated data for such attributes, without introducing substantial further novelty in methodology design. Hence, we focus on collision-free scene layout generation, which addresses a prevalent practical limitation in existing approaches, and leave the incorporation of stylistic and spatial controls with texts to future work."
>
> Joint conditioning, where text specifies furniture arrangements, is certainly an interesting direction, which we hope to pursue in the future.
>
> ## Ablation Study
> CF-GISS is a procedural pipeline that requires the output of each previous step to serve as the input for the next. Therefore, it does not naturally lend itself to ablation questions where components of the pipeline can be removed to study their necessity.
>
> We have included an ablation experiment in the revised manuscript (Appendix A.2), where the diffusion model is used to directly generate the segmentation map instead of our current RGB-based layout.
>
> ## Reasonable Collisions/Occlusions in Top Views
> Yes, we agree that there could be reasonable furniture overlaps, as discussed in Section 6 of our manuscript:
>
> > "CF-GISS currently does not accommodate natural object overlaps, such as placing a lamp on a table. However, this can be addressed by applying multiple layers of 2D layouts, which also effectively prevents vertical collisions, a feature we leave for future work."
>
> For example, we can assume that the second layer is always positioned on top of the first layer and potentially use the first layer as conditioning for generating the second. We believe this approach would effectively address scenarios involving natural object overlaps.
>
> For specific scenarios where parts of chairs can be placed under a table, a simpler solution is to treat the table and chairs as a single 3D model, as dining tables are typically paired with chairs for consistent styles and sizes. Meanwhile, we would like to highlight that most existing approaches completely fail at avoiding unreasonable collisions, as shown in Figure 5, making these methods utterly unsuitable for real-world applications. Addressing this major limitation is the primary focus of our paper.

---

### Author Response · Authors · 2024-11-21
**General Response**

We thank all the reviewers for their constructive feedback. We have provided a revised version of the manuscript with changes highlighted in blue, while unchanged text and captions remain in their original color. The most important changes include a new ablation experiment, in which we used diffusion to directly generate segmentation maps in response to Reviewer iEvZ’s comments, a report of inference efficiency in response to Reviewer wFS6’s comments, and an empirical comparison with PhyScene in response to Reviewer Vvb4’s comments, along with several minor corrections.

Please refer to our individual responses for addressing specific concerns.

---

### Author Response · Authors · 2024-11-25

Dear Reviewers,

Thank you again for your constructive feedback.

We hope our response has adequately addressed your concerns and clarified the questions raised in your comments. As the discussion period is coming to an end soon, we would greatly appreciate your final feedback. We look forward to hearing your thoughts and are open to further discussion should you have any additional questions or remaining concerns.

---

### Author Response · Authors · 2024-12-02
**Gentle reminder**

Dear Reviewers,

As today is the last day of the discussion period, we sincerely look forward to your final comments on our responses, and are happy to address any additional questions or remaining concerns.

Best regards,

Authors

---

### Meta-Review · Area_Chair_23Ev · 2024-12-22

**Metareview:**

The paper introduces CF-GISS, a novel framework for 3D indoor scene synthesis that employs a 2D diffusion model with intermediate representations (top-down views or RGB images) to generate collision-free scene layouts. The use of a 2D intermediate representation allows one to alleviate object collision issues in scene synthesis. It also supports condition scene generation with either images or textual descriptions. The paper also comes with a newly introduced dataset that is 1.4 times larger than previous one. The major criticisms of the paper are (i) the complexity of the proposed pipeline (without comprehensive analyses), (ii) insufficient experimental evaluations (missing several important baselines), and (iii) some components being unnecessary. The paper received very polarized reviews (2 reject, 1 accept, 1 marginally accept). While the ACs agree with the reviewers that tackling the issue of object intersection through the top-down image is neat, the ACs also agree that the paper could be improved further with more comprehensive experiments. After extensive discussions, the ACs decided to reject the paper. The authors are encouraged to incorporate feedbacks from the reviewers and re-submit the paper to a future venue.

**Additional Comments On Reviewer Discussion:**

The reviewers are primarily concerned about the over-claim of the paper, insufficient experimental comparisons, and lack of analysis. During the rebuttal phase, the authors have promised tone down their claim and have provided some additional comparisons. While these modifications indeed improve the quality of the papers and demonstrate positive signals, the reviewers are still concerned about some of the arguments the authors made. After extensive discussions, the ACs decided to reject the paper. The authors are encouraged to incorporate feedbacks from the reviewers, add more comprehensive experiments (eg, evaluating all methods on the proposed dataset) and re-submit the paper to a future venue.

---

### Decision · Program_Chairs · 2025-01-22

Reject